# Implementation of Sequence-Based Classification Methods for Motion Assessment and Recognition in a Traditional Chinese Sport (Baduanjin)

**DOI:** 10.3390/ijerph19031744

**Published:** 2022-02-03

**Authors:** Hai Li, Selina Khoo, Hwa Jen Yap

**Affiliations:** 1Teaching and Research Section of Physical Education, College of Sport, Neijiang Normal University, Neijiang 641112, China; 10000861@njtc.edu.cn; 2Centre for Sport and Exercise Sciences, University Malaya, Kuala Lumpur 50603, Malaysia; selina@um.edu.my; 3Department of Mechanical Engineering, Faculty of Engineering, University Malaya, Kuala Lumpur 50603, Malaysia

**Keywords:** inertial sensor measurement systems, motion accuracy, motion recognition, Baduanjin

## Abstract

This study aimed to assess the motion accuracy of Baduanjin and recognise the motions of Baduanjin based on sequence-based methods. Motion data of Baduanjin were measured by the inertial sensor measurement system (IMU). Fifty-four participants were recruited to capture motion data. Based on the motion data, various sequence-based methods, namely dynamic time warping (DTW) combined with classifiers, hidden Markov model (HMM), and recurrent neural networks (RNNs), were applied to assess motion accuracy and recognise the motions of Baduanjin. To assess motion accuracy, the scores for motion accuracies from teachers were used as the standard to train the models on the different sequence-based methods. The effectiveness of Baduanjin motion recognition with different sequence-based methods was verified. Among the methods, DTW + *k*-NN had the highest average accuracy (83.03%) and shortest average processing time (3.810 s) during assessing. In terms of motion reorganisation, three methods (DTW + *k*-NN, DTW + SVM, and HMM) had the highest accuracies (over 99%), which were not significantly different from each other. However, the processing time of DTW + *k*-NN was the shortest (3.823 s) compared to the other two methods. The results show that the motions of Baduanjin could be recognised, and the accuracy can be assessed through an appropriate sequence-based method with the motion data captured by IMU.

## 1. Introduction

Traditional Chinese sports are an essential part of physical education (PE) in universities in China. The official document issued by the Ministry of Education requires that the PE curriculum of all universities in China include traditional Chinese sports [1]. An official report in 2012 showed that 76.7% of universities in China had chosen Chinese martial arts as part of their PE curriculum. Although traditional Chinese sports have been incorporated into university PE curriculums, there have been some problems in implementation. The most serious problem is the high student–teacher ratio in Chinese universities. In 1999, the government in China decided to increase the number of students in universities, which led to a rapid rise in the number of students [2]. According to the latest official report [3], the number of students in universities in China hit a record high of 40.02 million in 2019. Of this number, 30.31 million were undergraduates, and the average number of students per university was 15,176. The insufficient number of PE teachers in universities in China and the number of students in PE courses exceeding the recommended numbers have resulted in a high student–teacher ratio [4]. Due to the large number of students in Chinese martial arts classes, teachers can neither provide individual guidance to students nor can they correct the errors in the motions of students [5]. Moreover, the Ministry of Education requires that the assessments in PE focus on formative assessment, instead of only using summative assessment [1]. However, PE teachers are unable to conduct formative assessment due to the high student–teacher ratio.

The application of motion capture (MoCap) in sports provides a way to solve the above problems in teaching traditional Chinese sports in PE in Chinese universities. In the past 10 years, researchers have developed various systems for different sports on motion data captured by MoCap [6,7]. These systems measure the motions of users in real-time on MoCap, analyse the captured motion data, detect errors in the motions, and give feedback to the users to help them recognise and correct their motions. In the study by Yamaoka et al. [7], a system-applied MoCap (Kinect device) for frisbee learners was developed. The 3D motion data of learners were captured using the Kinect device while throwing the flying disc. The system checked the positions in pre/exercise/post-exercise and gave information to learners. The results show that this system can effectively improve the motions of learners.

When considering the actual requirements of PE, it was found that not all MoCaps are suitable for PE courses. The photoelectric system (OMS) shows the highest measurement accuracy in MoCap, but the high cost is a barrier for using OMS in PE [8,9]. Electromagnetic systems (EMSs) are susceptible to interference in the electromagnetic environment [10]. The image processing system (IMS) has better accuracy than the EMS and an improved range compared to the OMS [11]. Many studies used a low-cost IMS (e.g., Kinect device) to capture the motion data to analyse motion in PE6 [12]. However, a low-cost IMS has some disadvantages: low accuracy, limited environment adaptability, and limited motion ranges led us to choose the inertial sensor measurement system (IMU) for this study.

Therefore, we aimed to develop a formative assessment system using IMU to assess the motion accuracy of Baduanjin to assist teachers and students identify errors in motions. Baduanjin is a popular traditional sport in China that consists of eight decomposition motions (Figure 1).

The formative assessment system needs to recognise and assess the motions of Baduanjin effectively. The motion sequences in Baduanjin are fixed (from Motion-1 to Motion-8) and cannot be changed. However, some students have problems practising the correct motions sequence while learning it. Therefore, the system needs to have the ability to recognise Baduanjin motions to evaluate whether the student is following the correct motion sequence. Recognising the motions of Baduanjin can be interpreted as a classification problem. During the classroom teaching session, teachers use the traditional manual grading method (vision) to assess the motion accuracy of Baduanjin. Therefore, the assessment of the motion accuracies of the Baduanjin motions is converted into a classification problem [14]. Commonly used classification methods can be divided into sample-based methods and sequence-based methods [14]. Few studies have applied sequence-based methods to assess and recognise motions of Baduanjin and other traditional Chinese sports, such as tai chi. Chen et al. [12] used DTW to assess the motions of tai chi. Sequence-based methods are used in this study, including dynamic time warping (DTW) combined with classifiers, hidden Markov model (HMM) [15], and recurrent neural networks (RNNs) [16,17].

## 2. Materials and Methods

This study is comprised of three steps. The first step was to recruit volunteers and use IMU to capture the motion data of Baduanjin. Teachers and students were recruited from a university in Southwest China to participate in the study. The motion data of Baduanjin of all participants were captured using IMU. The second step involved extracting motion data and keyframes. The raw motion data needed to be converted into quaternion format for data analysis and the extracted keyframes, to prevent data redundancy and reduce processing time. The last step was to verify the effectiveness of sequence-based methods for assessing motion accuracy and recognising the motions of Baduanjin. In order to assess the motion accuracy of Baduanjin, Baduanjin experts were invited to score the motion accuracies of the captured motions and then use the scores to train the models for the different sequence-based methods (Figure 2).

### 2.1. Recruiting Volunteers and Capturing Motion Data

Undergraduate students or teachers with no clinical/mental illnesses or physical disabilities from a university in Southwest China were invited to participate in this study. All participants were informed and understood the purpose and scope of the data in this study and any associated privacy risks, and they accepted the researcher’s commitment to protecting their privacy and the security of data. The study was conducted following the Declaration of Helsinki, and the protocol was approved by the University of Malaya Research Ethics Committee (UM.TNC2/UMREC-558).

The Baduanjin motions of the volunteers were captured with the commercial IMU “Perception Neuron 2.0”, developed by Noitom [18]. Perception Neuron 2.0 has 17 inertial sensing units with a 3-axis gyroscope, 3-axis accelerometer, and 3-axis magnetometer [19]. Sers et al. verified the effectiveness the commercial IMU in measuring accuracy [20]. The captured motion data were the “output” in the Biovision Hierarchical Structure [21] motion file through the Axis Neuron software developed by Noitom.

### 2.2. Motion Data Conversion and Keyframes Extraction

Fifty-six participants were recruited and divided into two groups. The first group comprised 20 students and a martial arts teacher. The teacher has bachelor and master’s degrees in traditional Chinese sports (martial arts) and more than 10 years of experience teaching Baduanjin. The IMU was used to capture the motions of these participants three times. The second group comprised 35 students, and IMU was used to measure the motions of these participants. Students were undergraduates without disabilities and no clinical or mental illnesses. Therefore, for each motion of Baduanjin, 98 “motion data” were captured. The entire dataset of motions included 98 × 8 = 784 motion data (760 motions of students and 24 motions of a teacher).

The BVH file output from Perception Neuron 2.0 used the rotation data on skeleton points represented by Euler angles. The data on Euler angles were converted to quaternions to avoid universal joint locking and singularity in the rotation data represented by Euler angles [22]. The quaternion is a four-dimensional super-complex number representing the three-dimensional vector space on the real number [23]. The symbol used to represent a quaternion is as follows:(1)q=[w,x,y,z]

In Equation (1), w is the scalar component, and x, y, z are the vectors. Set the rotation order of Euler angles as z, y, x, and set the rotation angles around the x, y, and z axes as α, β, γ, then the Euler angles can be converted to quaternions, as follows:(2)$$\begin{array}{cc} q & =\left[\begin{array}{l} w\\ x\\ y\\ z \end{array}\right]=\left[\begin{array}{l} \cos(\gamma /2)\\ 0\\ 0\\ \sin(\gamma /2) \end{array}\right]\left[\begin{array}{l} \cos(\beta /2)\\ 0\\ \sin(\beta /2)\\ 0 \end{array}\right]\left[\begin{array}{l} \cos(\alpha /2)\\ \sin(\alpha /2)\\ 0\\ 0 \end{array}\right]\\ & =\left[\begin{array}{c} \cos(\gamma /2)\cos(\beta /2)\cos(\alpha /2)+\sin(\gamma /2)\sin(\beta /2)\sin(\alpha /2)\\ \cos(\gamma /2)\cos(\beta /2)\sin(\alpha /2)-\sin(\gamma /2)\sin(\beta /2)\cos(\alpha /2)\\ \cos(\gamma /2)\sin(\beta /2)\cos(\alpha /2)+\sin(\gamma /2)\cos(\beta /2)\sin(\alpha /2)\\ \sin(\gamma /2)\cos(\beta /2)\sin(\alpha /2)-\cos(\gamma /2)\sin(\beta /2)\sin(\alpha /2) \end{array}\right] \end{array}$$

The motion data obtained from MoCap for a single motion is several thousand frames. Table 1 shows the mean duration and the mean number of frames of Baduanjin captured in the study.

Due to the limited storage space and bandwidth capacity available to users in actual applications, the large amount of collected motion data may limit its application [24]. The keyframe extraction technique, to extract a small number of representative keyframes from long motion sequences, has been widely used in motion analysis. Therefore, this technique was applied to extract keyframes to reduce the amount of motion data, to improve data storage and subsequent data analysis [24,25]. In order to extract the keyframes on the desired compression rate, a method that extracts the keyframes by a preset compression rate on k-means clustering was used in the study [26]. In this algorithm, Step 1 is to preset the value of k, which is the value of the preset keyframes. In this study, the value of k is determined by the preset compression rate of the extracted keyframes, as follows [27]:(3)k=C_rate×N

In Equation (3), k is the value of the preset keyframes, C_rate is the compression rate of the keyframes to be extracted, and N is the number of original frames. On the preset value of k, Step 2 uses k-means to extract k cluster centroids from the dataset of 3D coordinates ([x, y, z]) of the skeleton points of the original frame. In k-means, the distance is the applied Euclidean distance in this study. The skeleton model has 17 skeleton points. Therefore, a cluster centroid is composed of 51 (17 × 3) vectors. Based on these extracted cluster centroids, the keyframes were extracted by calculating the Euclidean distance of skeleton points between the cluster centroids and the original frames. The algorithm to extract the keyframe is shown in Figure 3.

The reconstruction error between the reconstructed frames and the original frames was calculated to evaluate the effects on extracting keyframes in this study [28]. Motion reconstruction is based on reconstructing non-keyframes by interpolating adjacent keyframes, thereby rebuilding the same number of frames as the original frames. The interpolation method is: p_1_ and p_2_ are the positions of the skeleton points of adjacent keyframes at time t_1_ and t_2_, calculation of p_t_ (representing the position of the non-key frame point at time t) as follows [29]:(4)$$\begin{array}{l} u(t)=\frac{t_{2}-t}{t_{2}-t_{1}},\\ p_{t}=u(t)p_{1}+(1-u(t))p_{2},\\ t_{1}<t<t_{2} \end{array}$$

After reconstructing the motion by interpolation, the reconstruction error is calculated as follows:(5)$$\begin{array}{l} Dis(p_{1}^{i}-p_{2}^{i})=\sum_{j=1}^{n}{\left\|p_{1,j}^{i}-p_{2,j}^{i}\right\|}^{2},\\ Error(m_{1},m_{2})=\frac{1}{N}\sum_{i=1}^{N}Dis(p_{1}^{i}-p_{2}^{i}) \end{array}$$

In Equation (5), n represents the number of skeleton points, in the study m = 17; $p_{1,j}^{i}$ is the coordinate of j skeleton point for i frame in original frames, and $p_{2,j}^{i}$ is the coordinate of the j skeleton point for the coordinate frames in reconstructed frames. $Dis(p_{1}^{i}-p_{2}^{i})$ represents the calculated distance of the human posture in the i frame between the original frames and reconstructed frames. m_1_ is the original frames, m_2_ is the reconstruction frames. N represents the number of frames for reconstructed frames or original frames. Error (m_1_, m_2_) is the reconstructed errors calculated by the distance of human posture.

Five different compression ratios (5%, 10%, 15%, 20%, 25%) are chosen to extract keyframes and calculate the reconstruction errors of the corresponding keyframes under the different compression ratios (Figure 4). This figure shows the average reconstruction error between the extracted keyframes on the corresponding compression rate and the original frames. The Y-axis represents the reconstruction error, and the X-axis represents the five compression ratios (5%, 10%, 15%, 20%, 25%) to the extracted keyframes. The reconstruction errors corresponding to the eight motions of Baduanjin under different compression ratios are represented by different symbols and line segments (as shown in the legend in the figure).

It can be seen from Figure 4 that as the compression rate decreases, the error of motion reconstruction increases. When the compression ratio increases from 15% to 25%, the reconstruction error does not change much. However, the reconstruction error increases significantly when the compression ratio is between 5% and 15%. Therefore, keyframes on the compression ratio (15%) are extracted to ensure that the compression ratio and reconstruction error are reasonable.

### 2.3. Traditional Manual Assessment of Baduanjin

Motion recognition can be regarded as a classification issue of time-varying data, where the test motion is matched with a pre-calibrated motion representing typical motion [30]. Therefore, assessing motion accuracy is similar to an issue for classifying motions. In this study, two martial arts teachers who have more than 10 years of experience teaching Baduanjin from a university in Southwest China were invited to assess the motion accuracy for each student’s motions. According to the grading scale of Baduanjin, the two teachers assess the motion accuracy of the students’ motions into three grades: fail, pass, and good. The Kendall correlation coefficient test is used to calculate the correlation between the scores results from two teachers to ensure the consistency of the evaluation.

Table 2 shows that the captured motions of students have three different grades (fail, pass, and good) except for Motion-8, with only two grades (good and pass). The two teachers explained that Motion-8 is a simple motion in Baduanjin, which the students easily pass (Figure 5). The Kendall coefficient values between the scores of the two teachers are higher than 0.7, indicating that the two teachers have a high degree of consistency in assessing motion accuracy in Baduanjin. Based on the motion data and the scores of the two teachers, the three sequence-based methods were applied to assess motion accuracy: dynamic time warping (DTW) combined with classifiers, hidden Markov model (HMM) and recurrent neural network (RNN).

#### 2.3.1. Dynamic Time Warping (DTW) Combined with Classifiers

In previous studies [12,31], one method assessed the motion accuracy on the differences between the motions of the students and the teacher. The duration of the captured motions was different. DTW, the method used for calculating differences for different time series was applied [32]. Although the difference between motions calculated by DTW can be used to recognise motions, it is difficult to classify the motions based on the difference to assess the motion accuracy without classifiers. Therefore, classifiers were used to classify the motions after calculating the difference between the students’ motions and the teacher’s motions through DTW. The specific steps are as follows:

Step 1: to calculate the minimum cumulative distance between corresponding skeleton points between two motions of students and teacher with DTW [13,32]. In order to prevent DTW from mismatching by excessive time warping, the global warping window was limited in this study, which was set to 10% of the entire window range: $\;0.1\times \max\left(n,m\right)$. n and m are, respectively, the frame length of the two motions. Since the warping path of each corresponding skeleton point of the two motions may be different, the average value of the cumulative distance of the warping path was used to indicate the differences between the points:
(6)$$dis(q_{stu}^{i},q_{tea}^{i})=\frac{DTW(q_{stu}^{i},q_{tea}^{i})}{length(path^{i})},i=1,2,\ldots ,n$$

In Equation (6), stu represents the students’ motions; tea represents the teacher’s motions; qi is the vectors of the quaternion of the i skeleton point in the two motions, and the total number of points is n. length(path^i^) is the length of the warping path on the i skeleton point.

Step 2: because the human posture is composed of multiple skeleton points, the overall difference between the two motions from students and the teacher was calculated as distance:
(7)$$D(m_{stu},m_{tea})=\frac{\sum_{i=1}^{n}dis(q_{stu}^{i},q_{tea}^{i})}{n}$$Step 3: after calculating the difference between the teacher’s motion and the student’s motion, the classifier is trained with the scores of the two teachers to classify motion-for-motion accuracy. A variety of classifiers were selected in the study, namely *k*-nearest neighbour (*k*-NN) [33], support vector machine (SVM) [34], naive Bayes (NB) [35], logistic regression [36], decision tree (DT) [37], and artificial neural network (ANN). Due to the diversity of ANNs, two commonly used ANNs: back propagation neural network (BPNN) and radial basis function neural network (RBFNN) [38], were chosen as classifiers (Figure 6).

BPNN is a multi-layer feedforward network trained according to the error backpropagation algorithm. In this study, BPNN was constructed with three layers. The first layer was the input layer. The number of neurons was equivalent to the dimension of the feature vectors; the second layer was the hidden layer, the tangent sigmoid equation was applied as the activation equation:(8)$$f(x)=\frac{2}{{\left(1+e^{-2x}\right)}^{-1}}$$

The third layer is the output layer.

The RBFNN is a kind of feedforward network trained using a supervised training algorithm, but the calculation and processing time is lower than that of BPNN. The main advantage of the RBFNN is that it has only one hidden layer and uses the radial basis equation as the activation equation [38]. Unlike BPNN, the Gaussian equation is used as the basic equation in RBFNN, as follows:(9)$$f(x)=\sum_{i=1}^{M}w_{i}\exp(\frac{{\left\|x-ci\right\|}^{2}}{d^{2}})$$

#### 2.3.2. Hidden Markov Model (HMM)

HMM is a sequence-based method that has been successfully applied in recognising human motion [39,40]. HMM is a double random process consisting of a hidden Markov chain and a set of explicit discrete probabilities or probability density functions [40]. HMM can be expressed as follows:(10)λ=[N,M,π,A,B]

In Equation (8), N presents the set of states in the HMM model, which is the set of possible states of the event. M is the set of possible observation events corresponding to each state, which is the observation set. π is the a priori probability of state occupancy. A is the state transition probability matrix. B is associated with the observations from the various states, which describes the probability of each observation event in each state. HMM can solve three kinds of problems: evaluation, decoding, and learning. In this study, recognising or classifying motions is an evaluation problem [41]. The method supposes the output probability in one observation sequence: O = o_1_, o_2_, ..., o_r_ of a given HMM(λ) is: $P(O\left|\lambda )\right.$. The state with the highest probability in $P(O\left|\lambda )\right.$, the result of recognition or classification. $P(O\left|\lambda )\right.$ can be calculated by the forward algorithm, a dynamic programming algorithm that uses a table to store intermediate values when establishing the probability of an observation sequence. α_t_(j) in the forward trellis represents the forward probability of the given HMM(λ), which the observed sequence o_1_, o_2_, ..., o_r_ at time t and the state is j:(11)$$\alpha_{t}(j)=P(o_{1},o_{2},\ldots ,o_{t},q_{t}=j\left|\lambda )\right.$$

α_t_(j) is calculated by superimposing all possible paths:(12)$$\alpha_{1}(j)=\pi_{j}b_{j}(o_{1}),j=(1,2,\ldots ,N)$$

Initialisation:(13)$$\alpha_{t}(j)=\left[\sum_{i=1}^{N}\alpha_{t-1}(i)a_{ij}\right]b_{j}(o_{t}),t=(1,2,\ldots ,T),\;j=(1,2,\ldots ,N)$$

In Equations (10) and (11), N represents the number of paths from t − 1 to t when the state at time t is j. Then:(14)$$P(O\left|\lambda )\right.=\sum_{i=1}^{N}\alpha_{T}(j)$$

Corresponding to the forward algorithm is the backward algorithm, which is used to train the model. β_t_(i) represents the backward probability of the given HMM(λ), which the observed sequence o_t+1_, o_t+2_, ..., o_T_ at time t and the state is i:(15)$$\beta_{t}(i)=P(o_{t+1},o_{t+2},\ldots ,o_{T}\left|q_{t}=i,\lambda )\right.$$

Initialisation:(16)$$\beta_{T}(i)=1i=(1,2,\ldots ,N)$$

Recursion:(17)$$\beta_{t}(i)=\sum_{j=1}^{N}a_{ij}b_{j}(o_{t+1})\beta_{t+1}(j),t=(1,2,\ldots ,T),\;i=(1,2,\ldots ,N)$$

#### 2.3.3. Recurrent Neural Network (RNN)

Motion data of humans is a type of sequence data on time. A RNN is a recursive neural network that takes sequence data as input, recurses in the evolution direction of the sequence, and all nodes (recurrent units) are connected in a chain network [42]. Trained by the end-to-end training method, RNN can solve sequence labelling problems with unknown input–output alignment [43]. However, traditional RNNs have the problem of vanishing gradients or exploding gradients. Vanishing gradients cause the parameters to be affected by short-term information, and the earlier information decays with the increase of the time steps. Exploding gradients cause long-term dependence, which refers to the state of the current system that may be affected by the previous state of the system that can lead to information overload [42]. Long short-term memory (LSTM), a type of RNN, is a neural network especially designed to reduce the problem of vanishing or exploding gradients by using gates to obtain relevant information and forget irrelevant information [44]. Therefore, LSTM was used as a method of RNN to assess the motion accuracy in this study.

Figure 7 shows the architecture of LSTM applied in the study. In the figure, X_t_ is input; h_t_ is hidden states; Y_t_ is output; U_t_ is the cell state, and Ϭ is the activation function. There are three gates in the hidden state of LSTM: forget, input, and output. The equations in Figure 7 are:(18)$$\begin{array}{l} f_{t}=\sigma (W_{f}•[h_{t-1},x_{t}]+b_{f})\;\;\;\;\;\;\;\;\;\;\;\;\;\;\;(\mathrm{Forget}\;\mathrm{gate})\\ i_{t}=o_{t}=\sigma (W_{o}•[h_{t-1},x_{t}]+b_{o})\;\;\;\;\;\;(\mathrm{Input}\;\mathrm{gate})\\ \sigma (W_{i}•[h_{t-1},x_{t}]+b_{i})\;\;\;\;\;\;\;\;\;\;\;\;\;\;\;\;\;\;\;\;\;\;\;\;(\mathrm{Output}\;\mathrm{gate})\;\\ {\tilde{C}}_{t}=\tanh(W_{C}•[h_{t-1},x_{t}]+b_{C})\\ C_{t}=f_{t}*C_{t-1}+i_{t}*{\tilde{C}}_{t}\\ h_{t}=o_{t}*\tanh(C_{t}) \end{array}$$

LSTM remembers and forgets the long sequence input data using the gates. However, LSTM also has problems: whether the gates included in the architecture of LSTM already provide good predictions and whether additional data training is needed to further improve the predictions [45]. Bidirectional LSTM (BiLSTM) is an improvement to this problem of LSTM, which enables additional training by training the input data twice. Based on the traditional LSTM for positive training on a time series from front to back, BiLSTM adds negative training on a time series from back to front [45]. The architecture of BiLSTM is shown in Figure 8.

Cho et al., in 2014, proposed a model similar to LSTM but with simpler calculations and implementations, namely the gated recurrent unit (GRU) [46]. Similar to LSTM, GRU is a positive training model based on time series [47]. However, the architecture of the hidden state in GRU is different from that of LSTM. The architecture of the hidden state in GRU applied in this study is shown in Figure 9.

In Figure 9, X_t_ is input; h_t_ is hidden states; Y_t_ is output, and δ is activation function. There are two gates in the hidden state of LSTM. The equations in Figure 9 are:(19)$$\begin{array}{l} r_{t}=\sigma (W_{r}•[h_{t-1},x_{t}]+b_{r})\;\;\;\;\;\;\;\;\;\;\;\;\;\;\;(\mathrm{Reset}\;\mathrm{gate})\\ z_{t}=\sigma (W_{z}•[h_{t-1},x_{t}]+b_{z})\;\;\;\;\;\;\;\;\;\;\;\;\;\;(\mathrm{Update}\;\mathrm{gate})\\ {\tilde{h}}_{t}=\tanh(W_{\tilde{h}}•[r_{t}*h_{t-1},x_{t}]+b_{\tilde{h}})\\ h_{t}=(1-z_{t})*h_{t-1}+z_{t}*{\tilde{h}}_{t} \end{array}$$

From Figure 7 to Figure 9, the hidden state of LSTM and GRU is similar, in that the output at time t is calculated using the hidden state at time t − 1 and the value of input at time t. In addition, the equation of the forget gate for LSTM is similar to the equation of the reset gate for GRU. The difference is that LSTM has three gates, whereas GRU has two gates. There is no output gate in GRU as in LSTM, which means GRU directly transmitted the memory to the next cell, but LSTM selects whether to transmit memory by the output gate.

#### 2.3.4. Evaluating the Effectiveness of Methods

In this study, cross-validation is applied to evaluate the effectiveness of different sequence-based methods for assessing motion accuracy. In cross-validation, the dataset is divided into a training set and a test set: (1) the training set is used to train the model and (2) the test set is used to test the trained model. The 10-fold cross-validation was used in the study. The confusion matrix and the assessment accuracy rate were used to express the effectiveness of different sequence-based methods. The confusion matrix shows the degree of confusion in the classification between different motions. The assessment accuracy is calculated as:(20)$$\mathrm{Assessment}\;\mathrm{Accuracy}=\frac{\mathrm{Number}\;\mathrm{of}\;\mathrm{correctly}\;\mathrm{classified}\;}{\mathrm{Overall}\;\mathrm{sample}\;\mathrm{size}}$$

## 3. Results

The assessment of motion accuracy used captured motion data and the traditional manual scoring results from the two teachers. With 10-fold cross-validation, the models on the sequence-based methods were trained. Using the scoring results from Teacher A as a standard, the accuracies of different sequence-based methods (on assessing motion accuracy) are shown in Table 3. Figure 10 shows an example of the confusion matrices for the sequence-based methods for Motion-1.

From Table 3, the results show that none of the methods had the highest accuracy for all eight motions of Baduanjin. The DTW + *k*-NN method had the highest accuracy in assessing five motions (Motion-1, Motion-2, Motion-5, Motion-7, and Motion-8). The highest accuracy for Motion-3 was the LSTM method, reaching 80.00%, the highest accuracy for Motion-4 was HMM, and BiLSTM with 90.53%. Lastly, the highest accuracy of Motion-6 was LSTM, with 84.21%. The highest average accuracy was the DTW + *k*-NN with 84.21%. From Figure 10, it can be seen from the confusion matrix of Motion-1 that, for most methods, the errors in classifying occur mainly in pass and fail motions. DTW + SVM and DTW + logistic regression classified all fail motions as pass motions.

When using the scoring results from Teacher B as a standard, the accuracy of different sequence-based methods on assessing motion accuracy is shown in Table 4. The confusion matrices for the sequence-based methods with Motion-1 is used as an example in Figure 11.

From Table 4, the results show that, in assessing motion accuracy, not one method had the highest accuracy for all eight motions of Baduanjin. However, DTW + *k*-NN had the highest accuracy in assessing five motions (Motion-1, Motion-2, Motion-4, Motion-6, and Motion-7), with its mean accuracy reaching 83.03%, which is the highest compared to other methods. The highest accuracy for Motion-3 was HMM, at 80.00%, and the highest accuracy for motion-5 was DTW + RBFNN at 86.32%. The highest accuracy of Motion-8 was the GRU at 88.42%. On the confusion matrix of Motion-1 (Figure 11), the result is similar to Figure 10 when using Teacher A as the standard. The classification errors occur mainly in pass and fail, especially DTW + SVM and DTW + logistic regression.

The processing time (training and classifying) of each method is presented in Table 5.

The results show that the processing time of DTW + *k-*NN was the shortest (3.810 s). The processing times of the three recurrent neural network methods (LSTM, BiLSTM, and GRU) were much longer, all exceeding 15 s.

### Recognising Motions of Baduanjin

For recognising motion with 10-fold cross-validation, the accuracy and processing time of the sequence-based methods are shown in Table 6. The confusion matrices of different sequence-based methods for motion recognition are shown in Figure 12.

The results show that, by using sequence-based methods to recognise the motions of Baduanjin, the DTW method combined with the classifier will improve their accuracy (except for DTW + RBFNN). It can be seen for the classifier *k*-NN and SVM, the accuracy of both methods exceeded 99.00% (DTW + *k*-NN reached 99.47% and DTW + SVM reached 99.61%). It was the highest accuracy among all verification methods tested in this paper. However, the accuracy of DTW + RBFNN was not satisfactory, only reaching 75.79%. Besides DTW + BFNN, the processing time of DTW combined with several classifiers was between 3.823 and 10.163 s. The processing time for DTW + *k*-NN was the minimum of all methods at 3.823 s. The accuracies of HMM and RNN (LSTM, BiLSTM, and GRU) also exceeded 96%. However, the processing times of these four methods were relatively high (between 61.144 and 239.190 s), especially LSTM, BiLSTM, and GRU. These three methods had much higher processing times compared to DTW combined with classifiers. BiLSTM, with the maximum processing time, was many times longer than DTW + *k*-NN with the minimum processing time.

#### The Chi-Square Test of the High Accuracy Methods on Recognising Motions

From the confusion matrices, it was found that the accuracy of DTW + *k*-NN, DTW + SVM, and HMM were all higher than 99%. The chi-square test on the number of correct and incorrect judged motions by the three methods were obtained, and the results are shown in Table 7 and Table 8 (calculation on SPSS 23.0).

From the results of the chi-square test, it is found that the three methods have no significant difference in the accuracy of recognising motions. Therefore, the effectiveness of the three methods on accuracy can be considered the same.

Second, the incorrect motions recognised by the three methods are not the same. For example, for DTW + SVM, 3 of the 706 motions were misclassified, including Motion-3, Motion-5, and Motion-7. All three motions were incorrectly recognised as Motion-4. However, for HMM, the problem was in recognising the motions in Motion-7 and Motion-8 as other motions.

## 4. Discussion

This research used IMU to capture motion data to assess motion accuracy and recognise Baduanjin motions that can be used to assist students in knowing their errors during practice. To assess motion accuracy, previous studies have suggested and verified that motion accuracy could be assessed by comparing the motions to be assessed with the standard motions [12,31]. For example, in a study involving assessing the motion accuracy of tai chi, DTW was used to calculate the differences between the motions of teachers and students to assess the motion accuracies of the students’ motions [12]. The correlation between the assessment results of this method and the assessment results between teachers and students reached 80%. This method assesses the motion accuracy by calculating the difference or similarity between the motions, in which this method needs a quantitative assessment for the accuracy of motions. However, in teaching and learning, the traditional assessment for the accuracy of motions is usually a manually graded assessment, a qualitative assessment, such as the existing method in the course of Baduanjin in universities. Therefore, in this study, the motion accuracy of Baduanjin was assessed as a classification problem.

In this study, three types of sequence-based methods were applied, and the traditional manual assessment results of teachers were used as the classification criteria to train the classifiers. It is proven that the results of some classifiers have the ability to approach teacher assessment scoring results. However, none of the classifiers have the highest accuracies in assessing the motion accuracies of all eight motions of Baduanjin. Among the classifiers, with Teacher A as the classification criteria, DTW + *k*-NN reached the highest accuracies in five motions (Motion-1, Motion-2, Motion-5, Motion-7, and Motion-8). However, there were some differences in the results using Teacher B as the classification criteria. Among the classifiers, DTW + *k*-NN reached the highest accuracies in five motions (Motion-1, Motion-2, Motion-4, Motion-6, and Motion-7). Although no selected classifier had the best accuracy for all eight motions of Baduanjin, the highest average accuracy was DTW + *k*-NN, which also had the lowest average processing time among the selected classifiers.

In addition, the motion recognition of Baduanjin was also studied. In actual applications, the motion can be recognised, and the motion accuracy assessed using the corresponding optimal classifier. The results of motion recognition show that the selected classifiers reached high accuracy (all above 90%), except for DTW + RBFNN (75.79%). The accuracy of the three classifiers (DTW + *k*-NN, DTW + SVM, and HMM) exceeded 99%. From the confusion matrix of the three classifiers and the chi-square test performed thereon, there was no significant difference in the accuracy of the three types of classifiers, indicating that the accuracies of the three classifiers for motion recognition under the existing experimental data were the same. Therefore, as the shortest processing time among the three classifiers, DTW + *k*-NN is concluded as the best choice for motion recognition of Baduanjin.

There is a limitation in this study: the captured motions used for assessing motion accuracies were relatively small, which could be why the accuracies of the three types of RNN (LSTM, BiLSTM, and GRU) were not high in assessing the motion accuracy. It is expected that by increasing the motion dataset in further research, the accuracy of RNN will be increased [48].

## 5. Conclusions

This study shows that, on the motion data captured by IMU, the sequence-based method to classify the Baduanjin motions can assess the motion accuracies of the Baduanjin motions and recognise the motions with high accuracy and short processing times. The methods verified in this study could be used to assess the motions of Baduanjin in PE. Based on the verified sequence-based method in this study, a formative assessment system for Baduanjin in PE can be developed.

## Figures and Tables

**Figure 1 ijerph-19-01744-f001:**
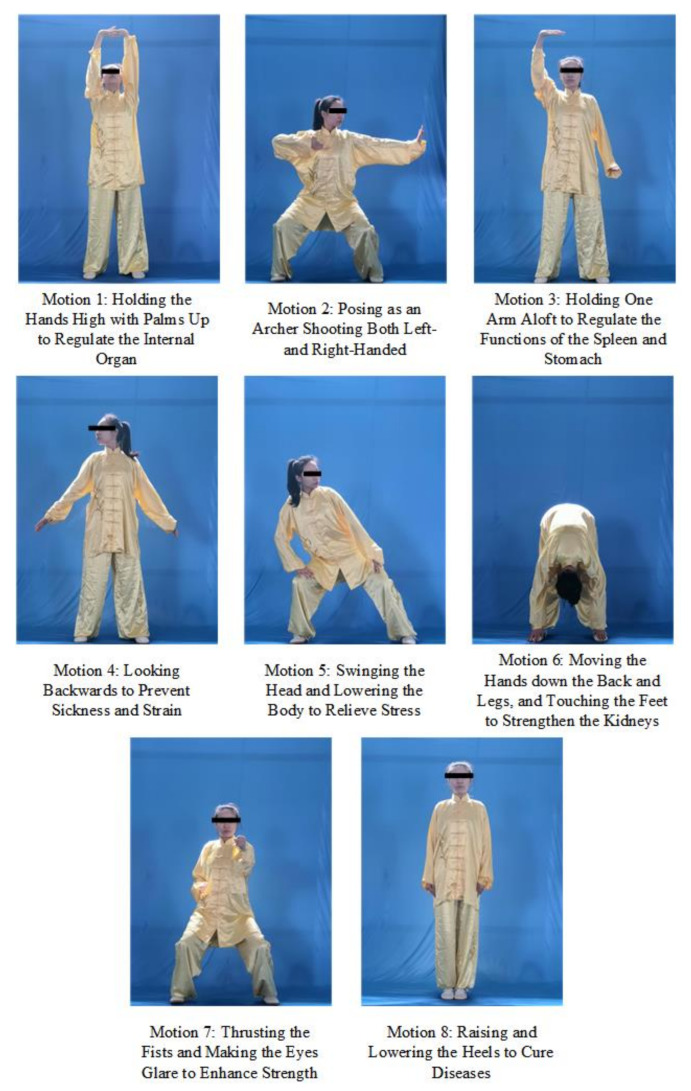
Eight standard motions of Baduanjin [13].

**Figure 2 ijerph-19-01744-f002:**
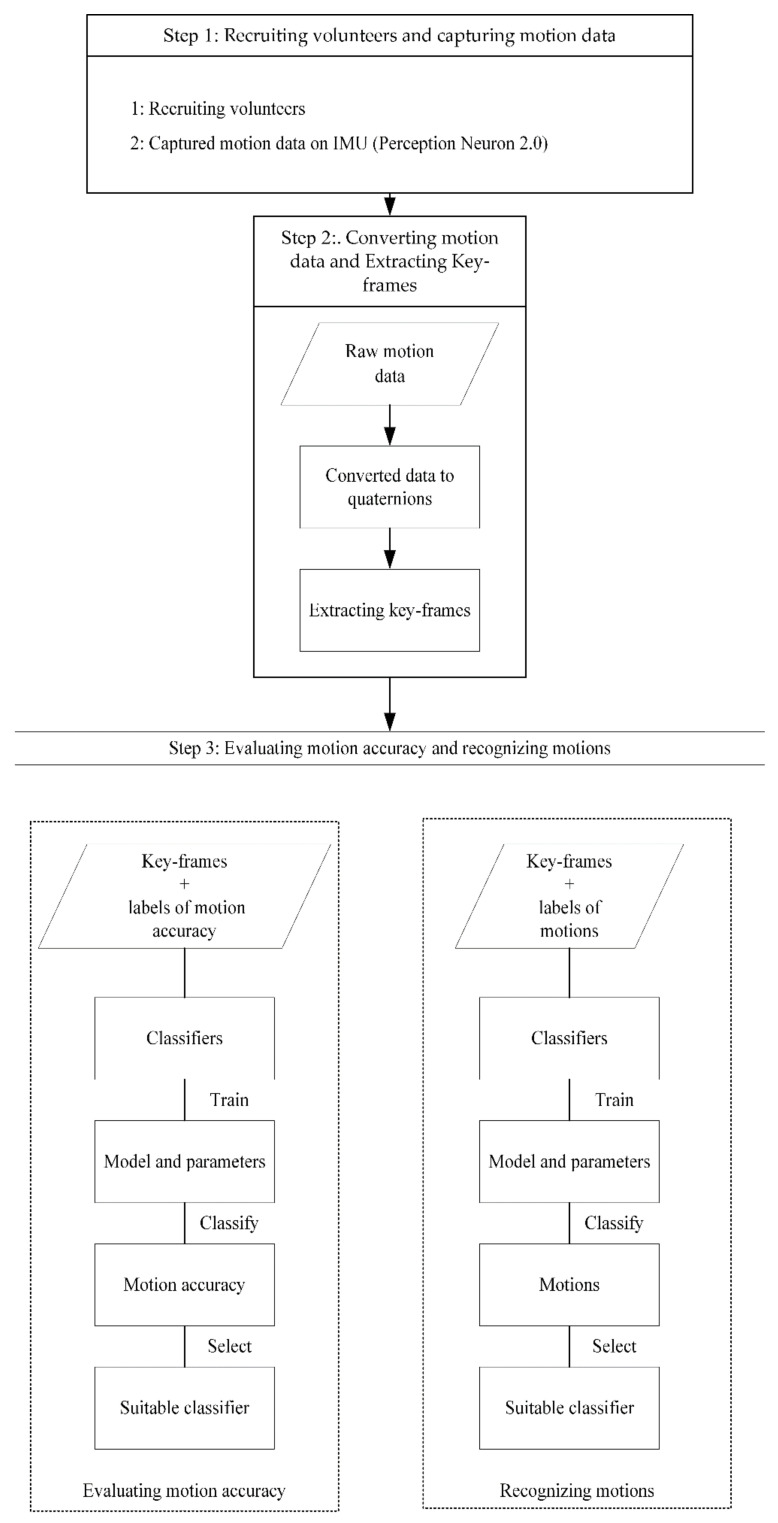
Flow diagram for the study.

**Figure 3 ijerph-19-01744-f003:**
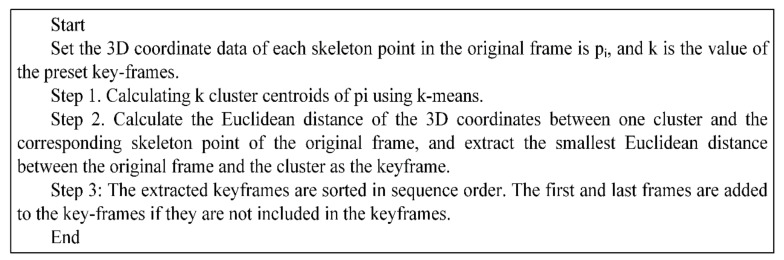
Algorithm to extract the keyframes from the original frames on the cluster centroids.

**Figure 4 ijerph-19-01744-f004:**
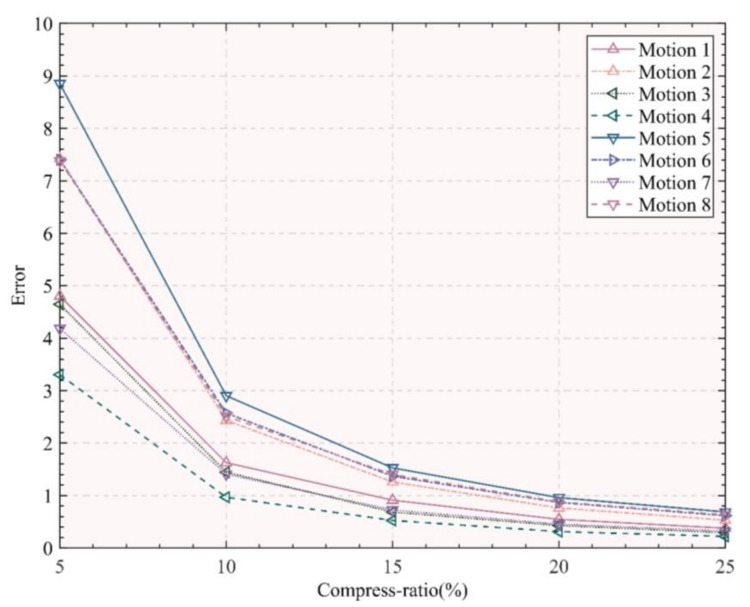
The average reconstruction errors of the corresponding keyframes under the different compression ratios.

**Figure 5 ijerph-19-01744-f005:**
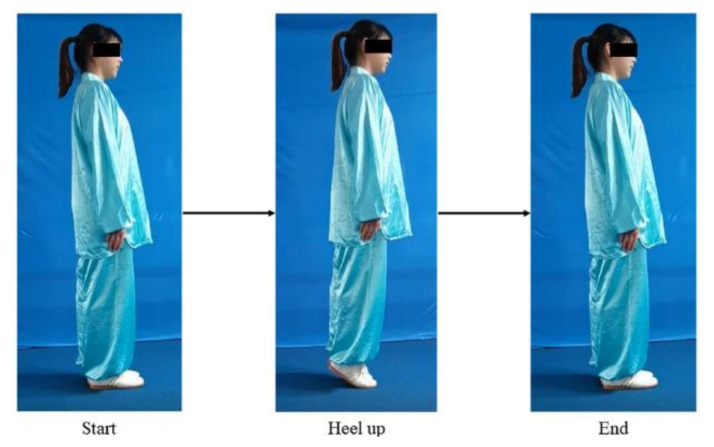
Motion-8.

**Figure 6 ijerph-19-01744-f006:**
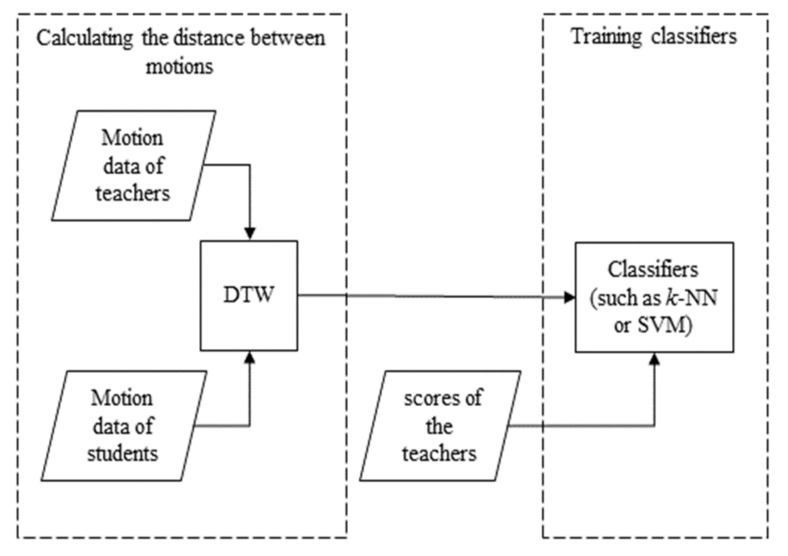
Flow diagram for DTW combined with classifiers.

**Figure 7 ijerph-19-01744-f007:**
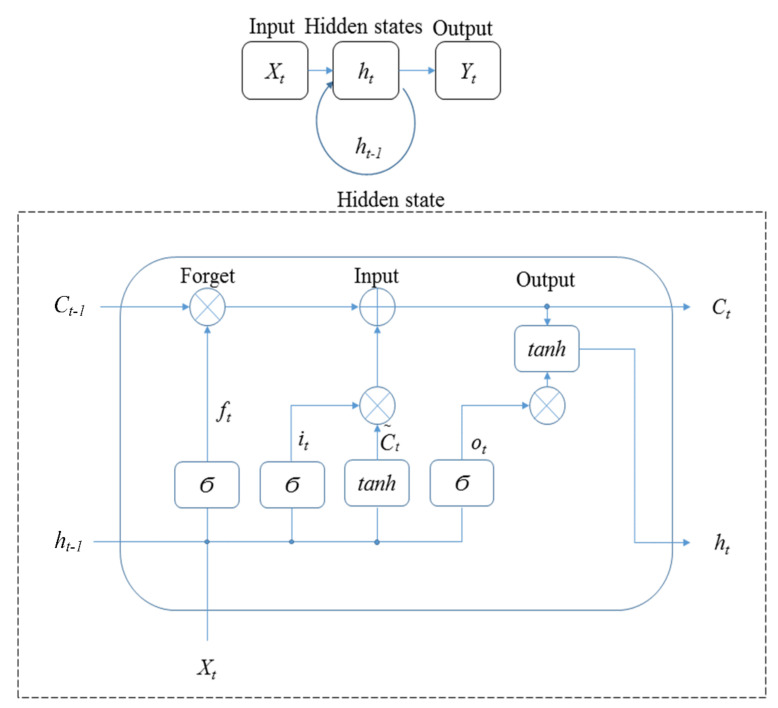
The architecture of LSTM.

**Figure 8 ijerph-19-01744-f008:**
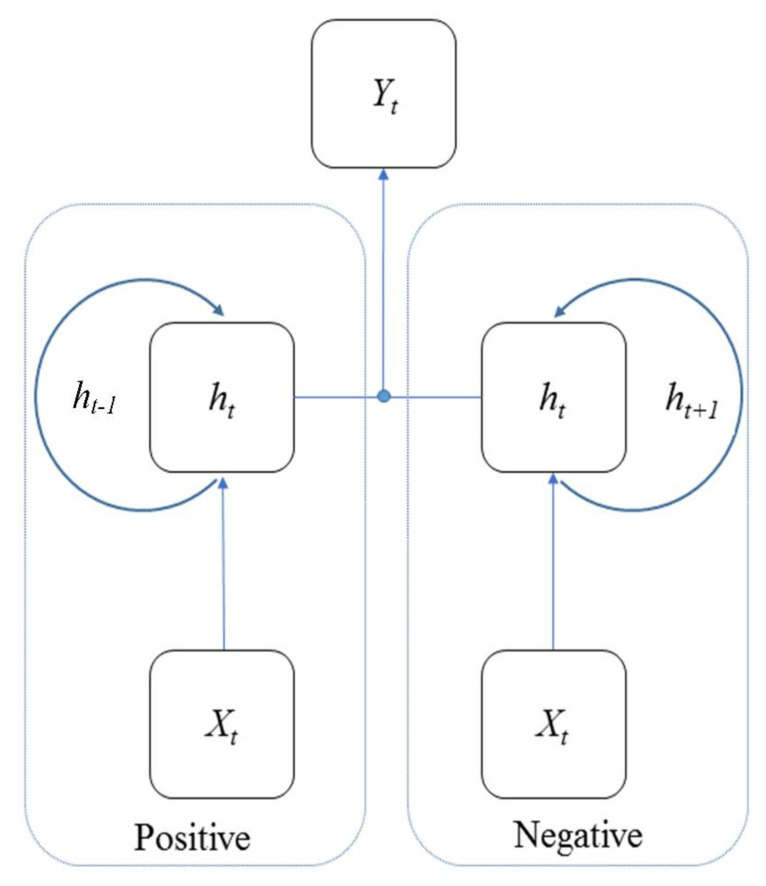
The architecture of BiLSTM. In the figure, the hidden units in BiLSTM are the same as in LSTM (see Figure 7).

**Figure 9 ijerph-19-01744-f009:**
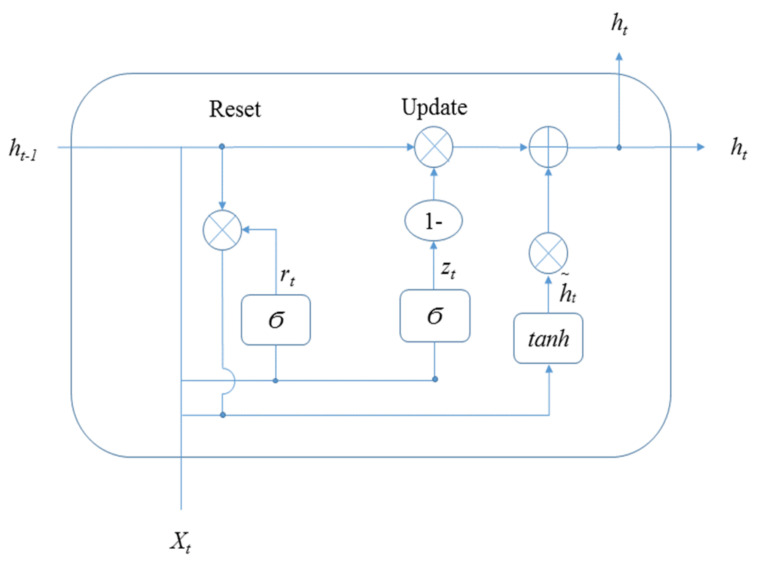
The typical architecture of the hidden state in GRU.

**Figure 10 ijerph-19-01744-f010:**
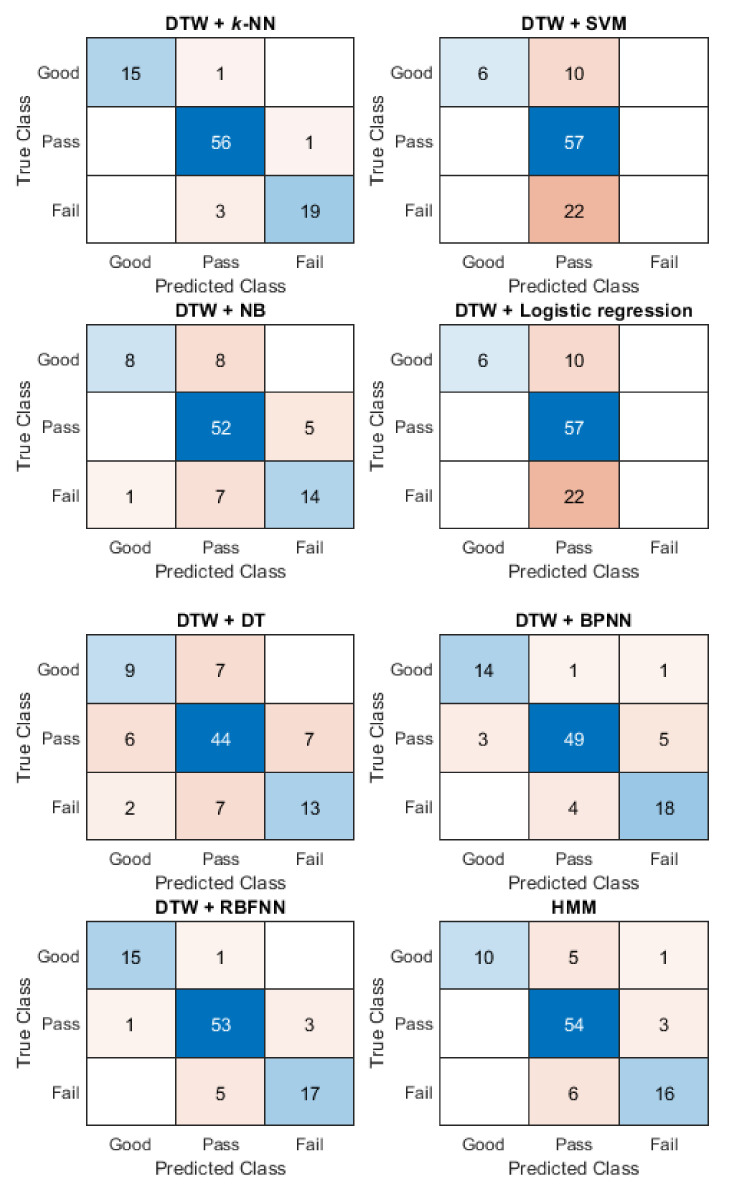
The confusion matrices of the different sequence-based methods for Motion-1 using the scoring results from Teacher A.

**Figure 11 ijerph-19-01744-f011:**
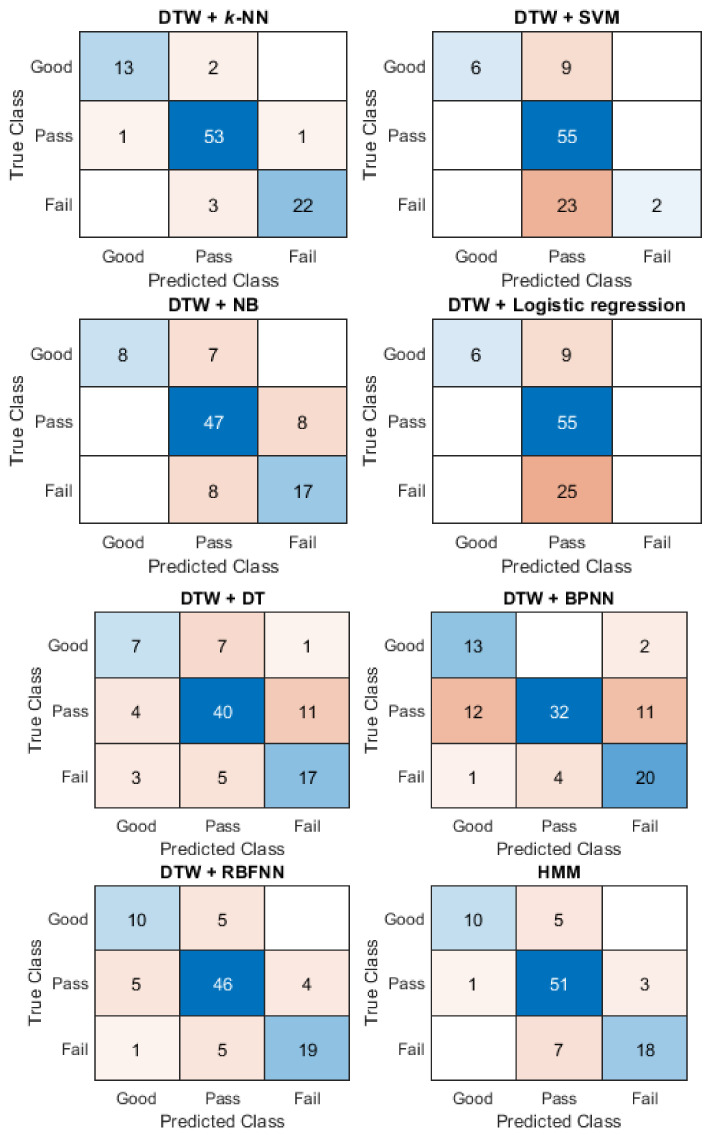
The confusion matrices of the different sequence-based methods for Motion-1 using the scoring result from Teacher B.

**Figure 12 ijerph-19-01744-f012:**
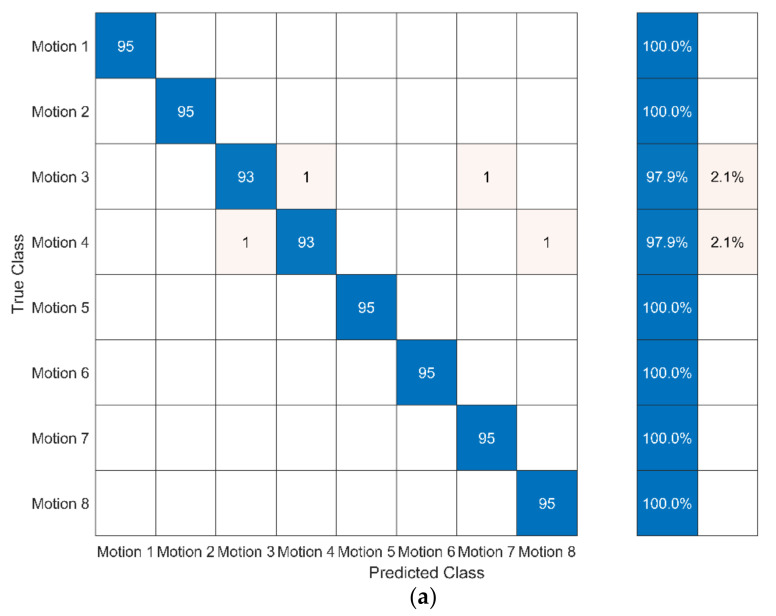
The confusion matrices for recognizing motion on different sequence-based methods: (**a**) DTW + *k*-NN; (**b**) DTW + SVM; (**c**) DTW + NB; (**d**) DTW + Logistic regression; I DTW + DT; (**f**) DTW + BPNN; (**g**) DTW + RBFNN; (**h**) HMM; (**i**) LSTM; (**j**) BiLSTM; (**k**) GRU.

**Table 1 ijerph-19-01744-t001:** The mean duration and the mean number of frames of Baduanjin.

Motion	Mean Duration (±SD) ^1^	Mean Number of Frames (±SD)
Motion 1	12.13 ± 2.80	1517 ± 350
Motion 2	21.72 ± 4.01	2715 ± 501
Motion 3	16.58 ± 3.78	2073 ± 472
Motion 4	15.09 ± 3.94	1887 ± 492
Motion 5	19.43 ± 4.67	2428 ± 583
Motion 6	16.40 ± 3.86	2050 ± 483
Motion 7	13.17 ± 3.50	1646 ± 438
Motion 8	2.94 ± 1.01	367 ± 126

^1^ second.

**Table 2 ijerph-19-01744-t002:** The scores of two teachers and Kendall values of the scores.

Motion	Teacher A			Teacher B			Kendall Value
	Good	Pass	Fail	Good	Pass	Fail	
Motion-1	16	57	22	15	55	25	0.941
Motion-2	21	53	21	24	50	21	0.882
Motion-3	26	58	11	23	49	23	0.831
Motion-4	22	58	15	19	49	27	0.824
Motion-5	20	57	18	20	56	19	0.838
Motion-6	23	55	17	20	54	21	0.907
Motion-7	29	59	7	26	62	7	0.944
Motion-8	61	34	0	61	34	0	0.862

**Table 3 ijerph-19-01744-t003:** The accuracies of assessing the motion accuracy of different sequence-based methods using the scoring results from Teacher A.

Methods	Accuracy (%)								
	Motion-1	Motion-2	Motion-3	Motion-4	Motion-5	Motion-6	Motion-7	Motion-8	Mean
DTW + *k*-NN	94.74 ^1^	86.32 ^1^	77.90	80.00	84.21 ^1^	77.90	87.37 ^1^	85.26 ^1^	84.21 ^1^
DTW + SVM	66.32	62.11	69.47	74.74	63.16	65.26	69.47	78.95	68.68
DTW + NB	77.90	72.63	74.74	84.21	65.26	70.53	70.53	74.74	73.82
DTW + Logistic regression	66.32	63.16	67.37	73.68	63.16	63.16	66.32	74.74	67.24
DTW + DT	69.47	63.16	82.11	70.53	67.37	68.42	74.74	69.47	70.66
DTW + BPNN	85.26	71.58	71.58	73.68	66.32	67.37	69.47	84.21	73.68
DTW + RBFNN	89.47	84.21	72.63	75.79	80.00	81.05	82.11	83.16	81.05
HMM	84.21	80.00	78.95	90.53 ^1^	76.84	78.95	83.16	77.90	81.32
LSTM	75.79	77.90	82.11 ^1^	84.21	72.63	84.21 ^1^	78.95	78.95	79.34
BiLSTM	84.21	80.00	78.95	90.53	76.84	78.95	83.16	77.90	81.32
GRU	80.00	75.79	67.37	83.16	74.74	81.05	82.11	72.63	77.11

^1^ The highest accuracy.

**Table 4 ijerph-19-01744-t004:** The accuracies of assessing the motion accuracy on different sequence-based methods using the scoring result from Teacher B.

Methods	Accuracy (%)								
	Motion-1	Motion-2	Motion-3	Motion-4	Motion-5	Motion-6	Motion-7	Motion-8	Mean
DTW + *k*-NN	92.63 ^1^	77.89 ^1^	77.90	80.00 ^1^	83.16	83.16 ^1^	86.32^1^	83.16	83.03 ^1^
DTW + SVM	66.31	60.00	69.47	69.47	66.32	65.26	72.63	77.90	68.42
DTW + NB	75.79	73.68	71.58	71.58	74.74	75.79	74.74	67.37	73.16
DTW + Logistic regression	64.21	61.05	61.05	68.42	61.05	62.11	70.53	71.58	65.00
DTW + DT	67.37	62.11	60.00	70.53	76.84	76.84	73.68	81.05	71.05
DTW + BPNN	68.42	62.11	66.32	65.26	66.32	54.74	71.58	78.95	66.71
DTW + RBFNN	78.95	74.74	76.84	62.11	86.32 ^1^	78.95	86.32 ^1^	83.16	78.42
HMM	83.16	73.68	80.00 ^1^	80.00^1^	77.90	76.84	82.11	85.26	79.87
LSTM	76.84	71.58	77.90	75.79	76.84	82.11	84.21	82.11	78.42
BiLSTM	82.11	75.79	78.95	74.74	76.84	82.11	83.16	85.26	79.87
GRU	76.84	67.37	69.47	71.58	75.79	78.95	77.90	88.42 ^1^	75.79

^1^ The highest accuracy.

**Table 5 ijerph-19-01744-t005:** Processing times (training and classifying) of the different sequence-based methods for assessing motion accuracy.

Methods	Processing Time (Seconds)
DTW + *k*-NN	3.810 ^1^
DTW + SVM	4.119
DTW + NB	4.057
DTW + Logistic regression	4.382
DTW + DT	3.947
DTW + BPNN	14.830
DTW + RBFNN	3.898
HMM	4.119
LSTM	14.132
BiLSTM	27.995
GRU	11.943

^1^ Minimum processing time.

**Table 6 ijerph-19-01744-t006:** Accuracy and processing time (training and classifying) of different sequence-based methods for recognising motion.

Methods	Accuracy (%)	Processing Time (seconds)
DTW + *k*-NN	99.47	3.823 ^2^
DTW + SVM	99.61 ^1^	6.909
DTW + NB	91.84	6.757
DTW + Logistic regression	94.21	10.163
DTW + DT	93.68	4.809
DTW + BPNN	91.05	24.665
DTW + RBFNN	75.79	5.439
HMM	99.08	61.144
LSTM	96.45	123.477
BiLSTM	97.37	239.190
GRU	97.50	106.513

^1^ The highest accuracy. ^2^ Minimum processing time.

**Table 7 ijerph-19-01744-t007:** The number of correct and incorrect judged motions by DTW + *k*-NN, DTW + SVM, and HMM.

Methods	Recognised Motions		Total
	Correct	Incorrect	
DTW + *k*-NN	756	4	760
DTW + SVM	757	3	760
HMM	753	7	760

**Table 8 ijerph-19-01744-t008:** The chi-square test results of DTW + *k*-NN, DTW + SVM, and HMM.

	Value	Degree of Freedom	Asymptotic Significance (2-Sided)	Exact Sig (2-Sided)
Pearson chi-square	1.869 ^1^	2	0.393	0.497
Likelihood ratio	1.804	2	0.406	0.497
Fisher’s exact test	1.722	--	--	0.497

^1^ 3 cells (50.0%) have expected count less than 5. The minimum expected count is 4.67.

## Data Availability

The data presented in this study are available on request from the corresponding author. The data are not publicly available due to the data involves the human body three-dimensional data of the participants.

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
