# Peer review of "Implementation of Sequence-Based Classification Methods for Motion Assessment and Recognition in a Traditional Chinese Sport (Baduanjin)"

_ijerph, 2022, doi:10.3390/ijerph19031744_

Round 1

Reviewer 1 Report

I would have better explained the biomechanics of movement and how the imus described these movements

Author Response

Dear Reviewer:

Thank you for giving us an opportunity to revise our manuscript. We appreciate you your constructive comments and suggestions on our manuscript entitled “Implementation of Sequence-based Classification Methods for Motion Assessment and Recognition in a Traditional Chinese Sport (Baduanjin)” (ijerph-1556363). The point-by-point response are as follows:

Point 1: I would have better explained the biomechanics of movement and how the imus described these movements.

Response: Thank you for the suggestion. We have added information of eight standard motions of Baduanjin in lines 76-79 as Figure 2

Reviewer 2 Report

I consider this article is relevant as it analyses a movement capture system that shows an improvement in the efficiency analysis of a physical activity such as Baduanjin, which can be extrapolated to other activities.

I would suggest the following improvements to this article:

In the introduction it is discussed that this study aimed to develop a system using the IMU to assess the accuracy of Baduanjin movements to help teachers and students identify movement errors. I suggest developing why that is important.

Method:

It should be made clear what kind of study is being conducted and also how the final sample of 33 people was obtained, specify the criteria for inclusion/exclusion from the study.

Although it is mentioned at the end of the article that informed consent was obtained, it should also appear in this section.

Line 433: The correlation between the results of the evaluation of this method and the results of the teacher-student evaluation reached 80%. This part should be better developed, as the accuracy level of the three classifiers (DTW + k-NN, DTW + SVM and HMM) is subsequently reported to be more than 99%.

Line 465: explain what implications the stated limitations of the study may have.

Add what benefits it might have to extrapolate this method to other kinds of physical activity.

Author Response

Dear Reviewer:

Thank you for giving us an opportunity to revise our manuscript. We appreciate you your constructive comments and suggestions on our manuscript entitled “Implementation of Sequence-based Classification Methods for Motion Assessment and Recognition in a Traditional Chinese Sport (Baduanjin)” (ijerph-1556363). The point-by-point response are as follows:

Point 1: In the introduction it is discussed that this study aimed to develop a system using the IMU to assess the accuracy of Baduanjin movements to help teachers and students identify movement errors. I suggest developing why that is important.

Response: Thank you for the suggestion. We have added the following:

Lines 46-50

Besides, the Ministry of Education requires that the assessments in PE focus on formative assessment instead of only using summative assessment [1]. However, PE teachers are unable to conduct formative assessment due to the high student-teacher ratio.

Lines 73-74

Therefore, we aimed to develop a formative assessment system using IMU to assess the motion accuracy of Baduanjin to assist teachers and students identify errors in motions.

Lines 486-489

The methods verified in this study could be used to assess the motions of Baduanjin in PE. Based on the verified sequence-based method in this study, a formative assessment system for Baduajin in PE can be developed.”

Point 2: It should be made clear what kind of study is being conducted and also how the final sample of 33 people was obtained, specify the criteria for inclusion/exclusion from the study.

Although it is mentioned at the end of the article that informed consent was obtained, it should also appear in this section.

Response: We have included the inclusion and exclusion criteria:

The teacher has bachelor and master degrees in traditional Chinese sports (martial arts) with more than 10 years of experience teaching Baduanjin.

Students were undergraduate students without a disability no clinical or mental illness.

Please see lines 123-124 and 127-128.

Point 3: Line 433: The correlation between the results of the evaluation of this method and the results of the teacher-student evaluation reached 80%.

This part should be better developed, as the accuracy level of the three classifiers (DTW + k-NN, DTW + SVM and HMM) is subsequently reported to be more than 99%.

Response:

  • The results of the teacher-student evaluation of 80% is the accuracy of sequence-based methods for assessing motion accuracy.
  • The accuracy level of three classifiers (DTW + k-NN, DTW + SVM and HMM) reported to be more than 99% is the accuracy of sequence-based methods for recognising motions.

Thus, the motion accuracy and recognising motions are two different functions developed for evaluating motions with two different results as (a) and (b)

Point 4: Line 465: explain what implications the stated limitations of the study may have.

Response: We have added the following content:

It is expected to increase motion dataset in further research, the accuracy of RNN will be increased [50].”

Please see lines 480-482

Reviewer 3 Report

I note that the study is based on three steps/blocks: 1) Recruitment to capture data through the IMU which is an inertial sensor measurement system, 2) Conversion/extraction of the movement data/photograms and 3) Traditional manual application to evaluate this Chinese sport.

The first two steps are very theoretical of applying formulas, and the third step which is the interesting one, is where the 3 methods of comparison and classification with their different types for the sequencing part and for the accuracy of movements in time.

In this evaluation part, two masters are taken to classify the results of the 8 movements, where he uses the DTW method for time distortion with different classifiers (kNN, SVM, NBayes, RLog, DT, etc; up to 7 different ones). Then he explains the techniques and compares results with Markov Models and Recurrent Neural Networks (3 different types) to see the effectiveness of accuracy. Already here the confusion matrices are observed for each master, first one and then the other, for the different types of methods, highlighting the most the kNN in both.

The same for accuracy in sequences is applied to processing time, showing confusion matrices and tables of results for the different types of methods. Then, it uses cross-validation (CV=10) for motion recognition and combining accuracy-time and shows the same. In both, kNN is the best in terms of shortest time (sec).

Finally, using SPSS and with the final confusion matrices to compare the three methods (k-NN, SVM and HMM) there is no difference between the two in terms of motion recognition accuracy. It can be seen that this may be due to the relatively small capture and he puts this as a limitation of the study. What was already intuitive and striking is the values of the matrices in the categories and this could be due to the small sample size used. I think that the sample is not sufficient and the sample size should be increased.

Comments:

  1. Clarify the sample used due to confusion. Point 2.2 (Materials and Methods) indicates 53 participants when the Abstract refers to 54 participants. However, if we continue reading the methods section where these 53 are divided into two groups (21+35), then we can see that the final sample is actually 56 participants including the teacher from the first group.

And so it is doubly verified that the sample is 56 with data extraction, where we have: G1 (21*3) is 63 + G2 is 35 = 98x8 (motion1to8) =784 (760 students + 24 teacher).

  1. Correct typo in Results Table 1. Delete last row of table due to erratum in Motion1 results repetition.

Author Response

Dear Reviewer:

Thank you for giving us an opportunity to revise our manuscript. We appreciate you your constructive comments and suggestions on our manuscript entitled “Implementation of Sequence-based Classification Methods for Motion Assessment and Recognition in a Traditional Chinese Sport (Baduanjin)” (ijerph-1556363). The point-by-point response are as follows:

Point 1: I think that the sample is not sufficient and the sample size should be increased.

Response: Similar studies have used comparable sample sizes:

  • Chen et al. built a Taichi motion dataset by capturing 150 motions from 30 students.
    • Chen, X.M.; Chen, Z.B.; Li, Y.; He, T.Y.; Hou, J.H.; Liu, S.; He, Y. (2019), ImmerTai: immersive motion learning in VR environments. Journal of Visual Communication and Image Representation 2019, 58, 416-427.
  • There were 5 participants in the study by Pham et al to research three-dimensional gesture.
    • Pham, M.T.; Moreau, R.; Boulanger, P. (2010), Three-Dimensional Gesture Comparison Using Curvature Analysis of Position and Orientation. In Proceedings of the 32nd Annual International Conference of the IEEE EMBS, Buenos Aires, Argentina, 31-Aug to 4-Sep 2010; 6345-6348.

Point 2: Clarify the sample used due to confusion. Point 2.2 (Materials and Methods) indicates 53 participants when the Abstract refers to 54 participants. However, if we continue reading the methods section where these 53 are divided into two groups (21+35), then we can see that the final sample is actually 56 participants including the teacher from the first group.

And so it is doubly verified that the sample is 56 with data extraction, where we have: G1 (21*3) is 63 + G2 is 35 = 98 x 8 (motion1 to 8) =784 (760 students + 24 teacher)

Response: Thank you for highlighting this. The correct sample size is 56 (Line 122). This has been corrected in the article.

Point 3: Correct typo in Results Table 1. Delete last row of table due to erratum in Motion1 results repetition.

Response: Thank you for highlighting this. We have deleted the last row in Table 1.

Reviewer 4 Report

Implementation of Sequence-based Classification Methods for Motion Assessment and Recognition in a Traditional Chinese Sport (Baduanjin)

This study aimed to develop a system using IMU to assess the motion accuracy of Baduanjin to assist teachers and students identify errors in motions. The paper has several strengths including the sophisticated analytical
approach. Yet, a number of issues need to be addressed.

Which conceptual gaps in the literature are addressed?

Would you please clarify the gap in the literature and what this study adds?

Descriptive information about ethical standards and informed consent should be presented. 

A statistical analyses section would help to understand the analyses better.

The sample of 54 participants seems very small. Is it too small for this
kind of advanced analyses? Are estimates reliable? Has an a priori power
analysis been conducted?

The sample size must be clarified. Different N are indicated 53, 54,
etc. Please clarify or correct.

The conceptual implications are not very clear. Can the authors better
specify in detail how and which current frameworks are advanced with
this study?

Likewise, the practical implications could be elaborated in more detail.

The authors may want to discuss in more detail the conceptual and applied implications of the findings.

The conclusion are not supported by the results.

The “Figure 1. The number of students in universities in China from 1978 to 2019 [3]” doesn’t make sense in the introduction.

The Figure 3. Algorithm to extract the key-frames from the original frames on the cluster centroids” doesn’t make sense.

Author Response

Dear Reviewer:

Thank you for giving us an opportunity to revise our manuscript. We appreciate you your constructive comments and suggestions on our manuscript entitled “Implementation of Sequence-based Classification Methods for Motion Assessment and Recognition in a Traditional Chinese Sport (Baduanjin)” (ijerph-1556363). The point-by-point response are as follows:

Point 1: Which conceptual gaps in the literature are addressed. Would you please clarify the gap in the literature and what this study adds?

Response: Few studies have applied sequence-based methods to assess and recognise motions of Baduanjin and other traditional Chinese sports, such as Taichi. Only Chen et al. [12] used DTW to assess the motions of Taichi. Please see lines: 89-91

Our research aimed to construct and verify the accuracy of sequence-based methods in assessing motions and recognising motions of Baduanjin.

  • Chen, X.M.; Chen, Z.B.; Li, Y.; He, T.Y.; Hou, J.H.; Liu, S.; He, Y. (2019), ImmerTai: immersive motion learning in VR environments. Journal of Visual Communication and Image Representation 2019, 58, 416-427.

Point 2: Descriptive information about ethical standards and informed consent should be presented.

Response: Add the content:

Lines: 109-112

Undergraduate students with no clinical/mental illness or physical disability from a university in Southwest China were invited to participate in this study. Participants read the information sheet that outlined the purpose and procedure of the study. We briefed participants on the procedures answered questions, and those who agreed to participate in the study were given the consent form to sign.

Point 3: A statistical analyses section would help to understand the analyses better.

Response: Thanks for your suggestion. We add the section as:

3.2.1. The Chi-square test of the high accuracy methods on recognising motions.”

Please see Line: 425

Point 4: The sample of 54 participants seems very small. Is it too small for this kind of advanced analyses? Are estimates reliable? Has an a priori power analysis been conducted?

Response: Similar studies have used comparable sample sizes:

  • Chen et al. built a Taichi motion dataset by capturing 150 motions from 30 students.
    • Chen, X.M.; Chen, Z.B.; Li, Y.; He, T.Y.; Hou, J.H.; Liu, S.; He, Y. (2019), ImmerTai: immersive motion learning in VR environments. Journal of Visual Communication and Image Representation 2019, 58, 416-427.
  • There were 5 participants in the study by Pham et al to research three-dimensional gesture.
    • Pham, M.T.; Moreau, R.; Boulanger, P. (2010), Three-Dimensional Gesture Comparison Using Curvature Analysis of Position and Orientation. In Proceedings of the 32nd Annual International Conference of the IEEE EMBS, Buenos Aires, Argentina, 31-Aug to 4-Sep 2010; 6345-6348.

Point 5: The sample size must be clarified. Different N are indicated 53, 54, etc. Please clarify or correct.

Response: Thank you for highlighting this. The correct sample size is 56. This has been corrected in the article. Please see line 122

Point 6: The conceptual implications are not very clear. Can the authors better specify in detail how and which current frameworks are advanced with this study?

Response: Thank you for the suggestion. We have added the following:

Lines 46-50

Besides, the Ministry of Education requires that the assessments in PE focus on formative assessment instead of only using summative assessment [1]. However, PE teachers are unable to conduct formative assessment due to the high student-teacher ratio.

Lines 73-74

Therefore, we aimed to develop a formative assessment system using IMU to assess the motion accuracy of Baduanjin to assist teachers and students identify errors in motions.

Lines 486-489

The methods verified in this study could be used to assess the motions of Baduanjin in PE. Based on the verified sequence-based method in this study, a formative assessment system for Baduajin in PE can be developed.

Point 7: Likewise, the practical implications could be elaborated in more detail.

Response: Add the content:

Thank you for the suggestion. We have added the following:

Lines 46-50

Besides, the Ministry of Education requires that the assessments in PE focus on formative assessment instead of only using summative assessment [1]. However, PE teachers are unable to conduct formative assessment due to the high student-teacher ratio.

Lines 73-74

Therefore, we aimed to develop a formative assessment system using IMU to assess the motion accuracy of Baduanjin to assist teachers and students identify errors in motions.

Lines 486-489

The methods verified in this study could be used to assess the motions of Baduanjin in PE. Based on the verified sequence-based method in this study, a formative assessment system for Baduajin in PE can be developed.

Point 8: The authors may want to discuss in more detail the conceptual and applied implications of the findings.

Response: We aim to develop a formative assessment system for Baduanjin in PE based on the developed and verified methods in this study.

We added the content as follow:

Lines: 73-74

Therefore, we aimed to develop a formative system using IMU to assess the motion accuracy of Baduanjin to assist teachers and students identify errors in motions.

Lines: 486-489

The methods verified in this study could be used to assess the motions of Baduanjin in PE. Based on the verified sequence-based method in this study, a formative assessment system for Baduajin in PE can be developed.

Point 9: The conclusion are not supported by the results.

Response: We added the content in conclusion:

Lines: 486-489

The methods verified in this study could be used to assess the motions of Baduanjin in PE. Based on the verified sequence-based method in this study, a formative assessment system for Baduajin in PE can be developed.

Point 10: The “Figure 1. The number of students in universities in China from 1978 to 2019 [3]” doesn’t make sense in the introduction.

Response: Thank you for the suggestion. We deleted this Figure.

Point 11: The Figure 3. Algorithm to extract the key-frames from the original frames on the cluster centroids” doesn’t make sense.

Response: Extracting the key-frames from the original frames on the cluster centroids is an algorithm based on k-means. Figure 3 presents the computer programming of this algorithm.

Round 2

Reviewer 3 Report

Thank you for your response. Despite what you say, with such a small sample size and subdividing it into two groups, the data aren't quite normal and the conclusions may be biased by the responses and data used. I believe that if you could increase the sample size it would improve the quality of the results and the conclusions drawn.

Author Response

Dear Reviewer:

Thank you for giving us an opportunity to revise our manuscript. We appreciate you your constructive comments and suggestions on our manuscript entitled “Implementation of Sequence-based Classification Methods for Motion Assessment and Recognition in a Traditional Chinese Sport (Baduanjin)” (ijerph-1556363). The point-by-point response are as follows:

Point 1: I believe that if you could increase the sample size it would improve the quality of the results and the conclusions drawn.

Response:

Thank you for your suggestion.

We used cross-validation (10-fold cross-validation) in training models to ensure a certain degree of fit to the data outside the training dataset. When training the model, the required verification index reached the plateau period. We will expand the research on selected methods (high accuracy and suitable processing time) for future development of further training and optimisation to obtain the best accuracy.

Reviewer 4 Report

  • Institutional Review Board Statement is a critical consideration inResearch in Humans. Authors should follow the ethical standards of the Declaration of Helsinki for Medical Research in Humans (2013) and Oviedo Convention (1997).
  • A statistical analyses section is lacking. The authors must present in detail the steps done in the statistical analyses. It will allow the replication of these studies by other research teams. 
  • Concerning sample size, it is not acceptable the author's response "Similar studies have used comparable sample sizes:". Please provide accurate calculations of sample size providing power analysis. 
  • General questions, "conclusion are not supported by the results" and discuss in more detail the conceptual and applied implications of the findings"were not answered appropriately. 

Author Response

Dear Reviewer:

Thank you for giving us an opportunity to revise our manuscript. We appreciate you your constructive comments and suggestions on our manuscript entitled “Implementation of Sequence-based Classification Methods for Motion Assessment and Recognition in a Traditional Chinese Sport (Baduanjin)” (ijerph-1556363). The point-by-point response are as follows:

Point 1: Institutional Review Board Statement is a critical consideration in Research in Humans. Authors should follow the ethical standards of the Declaration of Helsinki for Medical Research in Humans (2013) and Oviedo Convention (1997).

Response: The study was approved by the University of Malaya Research Ethics Committee (UM.TNC2/UMREC–558).

The Universiti Malaya Research Ethics Committee has and approved the ethics application according to the ethical standards stated on their website:

https://umresearch.um.edu.my/university-of-malaya-research-ethics-committee-umrec

Point 2: A statistical analyses section is lacking. The authors must present in detail the steps done in the statistical analyses. It will allow the replication of these studies by other research teams.

Response: This study aimed to establish and identify artificial intelligence (AI) algorithms, rather than statistical methods, to assess the motion accuracy and recognise the motions of Baduanjin. Thus,

  • Using the Matlab software (as used in this paper), other researchers would be able to reproduce and replicate the study using AI algorithms stated in this paper.
  • The chi-square test (the statistical method in this paper) can be directly reproduced through SPSS.

Point 3: Concerning sample size, it is not acceptable the author’s response “Similar studies have used comparable sample sizes:”. Please provide accurate calculations of sample size providing power analysis.

Response: The AI algorithm is based on supervised classification. we used cross-validation (10-fold cross-validation) in training models to ensure a certain degree of fit to the data outside the training dataset. When training the model, the required verification index reached the plateau period. Therefore, the sample size is for this study is acceptable. Furthermore, this study aimed to identify suitable AI algorithms (sequence-based) for assessing the motion accuracy and recognising motions of Baduanjin. Based on the results presented in the study (AI with high accuracy), the sample size can be expanded in future research to optimise and improve accuracy.

Point 4: General questions, “conclusion are not supported by the results”, and discuss in more detail the conceptual and applied implications of the findings” were not answered appropriately.

Response: The conclusions are supported by the results as follows:

  • This study shows that on the motion data captured by IMU, the sequence-based method to classify the Baduanjin motions could assess the motion accuracy of the Baduanjin action and recognise the motions with high accuracy and short processing time.” (Lines: 487-488)
  • For assessing motion accuracy, the highest average accuracy is the DTW + k-NN with 84.21% on the labels of Teacher A (please see Table 3) and with 83. 03% on the labels of Teacher B (Please see Table 4). The processing time of DTW + k-NN is 3.810 seconds (please see Table 5)
  • For recognising the motions, three methods (DTW + k-NN, DTW + SVM and HMM) in the paper are over 99% accurate. The processing time of DTW + k-NN (3.810 seconds) and DTW + SVM (4.119 seconds) is not long (please see Table 6).
  • The methods verified in this study could be used to assess the motions of Baduanjin in PE.” (Lines: 489-490)
  • DTW + k-NN can be used to assess motion accuracy and recognise motions of Baduanjin.
  • Based on the verified sequence-based method in this study, a formative assessment system for Baduajin in physical education can be developed.
    • “The methods verified in this study could be used to assess the motions of Baduanjin in PE” (Lines: 490-491) is a deduction drawn from conclusions 4.1 and 4.2.
